# A monoclonal antibody raised against *Acinetobacter baumannii* capsular carbohydrate exhibits cross-species *in vitro* binding against *Pseudomonas aeruginosa*

Matthew Slarve[1], Hadley Jaramillo[1], Brian Luna[1]*, Brad Spellberg[2]

**1** Department of Immunology and Immune Therapeutics, Keck School of Medicine at USC, Los Angeles, California, United States of America, **2** Los Angeles General Medical Center, Los Angeles, California, United States of America

* brianlun@usc.edu

## Abstract

The emergence of extensively drug-resistant bacterial pathogens such as *Acinetobacter baumannii* and *Pseudomonas aeruginosa* necessitates the innovation of non-small molecule therapies. Monoclonal antibodies represent a promising non-small molecule option. By exploiting shared or similar antigenic structures on these pathogens, it may be possible to develop therapeutic antibodies that target both bacterial species. Here we demonstrate that a previously developed monoclonal antibody targeting *A. baumannii* (MAb10) has cross-species reactivity to *P. aeruginosa* in *in vitro* assays. Despite MAb10's distinct efficacy against *A. baumannii*, no efficacy was detected against *P. aeruginosa* in our mouse models of infection. Nevertheless, the unique carbohydrate structures targeted by MAb10 and shared by numerous bacterial species may underscore a possible strategy for developing therapeutic antibodies that can treat infections caused by multiple pathogens.

## Introduction

With the rise of antibiotic resistance, extensively drug resistant bacteria such as the ESKAPE pathogens (*Enterococcus faecium, Staphylococcus aureus, Klebsiella pneumoniae, Acinetobacter baumannii, Pseudomonas aeruginosa* and *Enterobacter*) have become subjects of critical concern for the development of novel therapies, with monoclonal antibodies (MAbs) emerging as highly desirable alternative to traditional small molecule antibiotics [1–3]. Our lab has long sought to develop MAbs against *A. baumannii*, and have found success using a target-agnostic discovery approach that prioritizes antibody activity with a broad panel of clinical isolates over specifically selected antigens [4–9].

Using this approach, we previously selected a MAb known as MAb10, which had *in vitro* binding and *in vivo* efficacy against *A. baumannii* isolates with specific capsule

**Data availability statement:** All relevant data are within the paper and its Supporting information files.

**Funding:** This work was funded by National Institute of Allergy and Infectious Diseases (NIAID) grant 2R01 AI130060 to Brad Spellberg. The funders had no role in study design, data collection and analysis, decision to publish, or preparation of the manuscript.

**Competing interests:** The authors have declared that no competing interests exist. This does not alter our adherence to PLOS ONE policies on sharing data and materials.

carbohydrate structures (K-antigens) [9]. Hence, we sought to determine what common motifs these K-antigens held, which were absent in K-antigens that MAb10 did not react with. Of the numerous K-antigens MAb10 binds to, only the structures of K2, K6, K42, and K46 were characterized in literature [10–13]. A common thread between these structures was the presence of pseudaminic acid (Pse) – a sugar belonging to the nonulosonic acid family of sugars, and only known to be expressed by pathogenic bacteria [14]. Working under the hypothesis that MAb10 reacts with Pse, we sought to evaluate whether it would have activity or protective efficacy against the bacteria that is this sugar's namesake: the Gram-negative ESKAPE pathogen *P. aeruginosa*, some serotypes of which express Pse in the variable region of LPS (O-antigens) [15]. Here, we demonstrate that MAb10 – which was produced by immunizing mice with *A. baumannii* alone – also has binding against *P. aeruginosa* isolates that have Pse in their O-antigens but lacks therapeutic efficacy.

## Results

Following the observation that the *A. baumannii* K-antigens that MAb10 binds to all contain Pse, which was initially isolated from *P. aeruginosa* O-antigens, we sought to compare the overall carbohydrate structures this sugar appears in between these organisms (Table 1) [16]. We have previously published data showing that MAb10 has robust binding against *A. baumannii* strains that express the four K-antigens shown in Table 1 [9]. Given that MAb10 binding is robust to the structural variations between these four K-antigens, including the placement of Pse within the branching carbohydrate structure, α/β isomers of Pse, modifications on the Pse monomer itself, and dramatically different neighboring sugars in the polysaccharide, we hypothesized that MAb10 would be able to bind to the carbohydrate structures of *P. aeruginosa* O7 and O9 antigens.

This binding was first demonstrated via flow cytometry – the O7 expressing strain 1224 had low but marked binding with 10 µg/mL MAb10 producing only 5.3% binding, and 50 µg/mL MAb10 elevating the reaction to 20.1% binding (Fig 1A), with percent binding calculated as the percentage of events in the MAb10 treated condition that were brighter than the events in the control condition. MAb10 had far superior activity with O9 expressing strains, including a clinical isolate (Fig 1B), with which 10 µg/mL MAb10 was sufficient to produce 94.7% binding, and strain 33356, for which 10 µg/mL MAb10 produced nearly 100% binding (Fig 1C). Additionally, MAb10 was tested against PA-01, an O5 serotype reference strain, which does not contain pseudaminic acid in its O-antigen [17,18]. No appreciable binding was observed with either 10 or 50 µg/mL MAb10, supporting the hypothesis that MAb10 may bind pseudaminic acid (Fig 1D). To determine how prevalent MAb10 reactivity would be with a broader sample of clinical *P. aeruginosa* isolates, we screened our full library of 118 isolates for MAb10 binding via flow cytometry and found that 25 strains (21.1%) reacted with MAb10 (Fig 1E).

We had previously demonstrated that MAb10 drives macrophage opsonophagocytosis of *A. baumannii in vitro* [9]. To determine if MAb10 would also promote the opsonophagocytosis of *P. aeruginosa*, we performed an opsonophagocytosis assay

**Table 1. MAb10-binding carbohydrates in *A. baumannii* are similar to O-antigens expressed by some serotypes of *P. aeruginosa*.** Capsule carbohydrate structures for *A. baumannii* K-antigens with known binding to MAb10 were obtained via literature search and compared to published structures for *P. aeruginosa* O-antigens. Pse, as shown in bolded and italicized font, is expressed in all *A. baumannii* K-antigens bound by MAb10, as well as *P. aeruginosa* O-antigens O7 and O9. Brackets enclose the repeat unit structures, while sugars outside of the bracket indicate branching sugars.

| Species | Type | Carbohydrate Structure | Ref |
|---|---|---|---|
| *A. baumannii* | K2 | *α-Pse5ac7Ac*-(2→6)-ß-D-Glcp-(1→6)-[→3)-ß-D-Galp-(1→3)-ß-D-GalpNAc-(1→] | [10] |
| *A. baumannii* | K6 | [→4)-*ß-Pse5NAc7Ac*-(2→6)-ß-D-Galp-(1→3)-ß-D-GalpNAc-(1→] | [11] |
| *A. baumannii* | K42 | *α-Pse5ac7Ac*-(2→4)-[→3)-ß-D-Ribp-(1→3)-ß-D-GalpNAc-(1→] | [12] |
| *A. baumannii* | K46 | *ß-Psep5Ac7Ac4Ac*-(2→6)-[→3)-α-D-Galp-(1→6)-α-D-GlcpNAc-(1→3)-ß-D-GalpNAc-(1→] | [13] |
| *P. aeruginosa* | O7 | [→3)-ß-D-FucNAC-(1→4)-*α-Pse5NAc7NFm*-(2→4)-ß-D-Xyl-(1→] | [16] |
| *P. aeruginosa* | O9 | [→3)-ß-D-QuiNAc-(1→)-(R)-CH3CHCH2CO-(→7)-*ß-Pse5NAc7N*-(2→4)-α-D-FucNAc-(1→] | [16] |

with MAb10 against strain 33356. MAb10 modestly elevated uptake with a median of 20.5 bacterial cells engulfed by macrophages compared to a median of 15 bacterial cells in the control group (Mann-Whitney U Test, p-value = 0.04761) (Fig 2A).

Given the clinical success of an existing MAb against *P. aeruginosa* O-antigen, We then sought to evaluate the protective efficacy of MAb10 *in vivo* using pulmonary and bloodstream infection models [19]. We had previously established that MAb10 was highly efficacious against lethal infection by *A. baumannii* with a dose as low as 1 µg. However, given the modest uptake shown in the opsonophagocytosis assay against *P. aeruginosa*, higher doses of MAb10 were used for the *in vivo* experiments. Male BAlb/C mice were infected intravenous (Fig 2B) or by an oral aspiration route (Fig 2C) with strain 33356 which has been used by previous investigators in infection models with BAlb/C mice [20]. Immediately following infection, mice were given intravenous treatment with a single dose of 150 µg or 300 µg MAb10, or isotype control. Mice were followed for seven days post-challenge. Neither infection model showed a benefit of MAb10 treatment against strain 33356, with all MAb10 treated groups having comparable survival to their respective control group.

## Discussion

We have previously shown that MAb10 broadly binds diverse *A. baumannii* isolates, and has excellent *in vivo* activity in our mouse model of bloodstream infection [9]. Here we explored its ability to bind and effectively treat infections caused by *P. aeruginosa*, given the observation that MAb10 has considerable *in vitro* binding potential with both pathogens. This may be because of oligosaccharides containing pseudaminic acid presenting as similar surface targets that are shared between both organisms, though further analysis would be required to rule out the possibility of other motifs mediating this binding. However, despite the measurable *in vitro* binding observed via flow cytometry, no *in vivo* efficacy was detected.

These results underscore several important points. First, it may be possible to find common antigens across different species and genera of pathological bacteria. For example, the PSE sugar has only been found to be expressed by pathogenic bacteria, having first been discovered in *P. aeruginosa*, followed by *Shigella boydii*, *Campylobacter jejuni, Helicobacter pylori, A. baumannii*, and enteroinvasive and enterotoxigenic strains of *Escherichia coli* [10,14,21–24]. Second, however, is that merely binding to target antigens is not sufficient to ensure clinical efficacy *in vivo*. We found that MAb 10 bound robustly *in vitro* to strains of *P. aeruginosa but* did not mediate protective efficacy *in vivo*. Although MAbs against the O-antigen of *P. aeruginosa* have seen success not only in mouse models but also clinical trials [19], and MAb10 has excellent potency against *A. baumannii* [9], the therapeutic effect is not one-size-fits-all, and protection against one pathogen does not necessarily equate to protection against another.

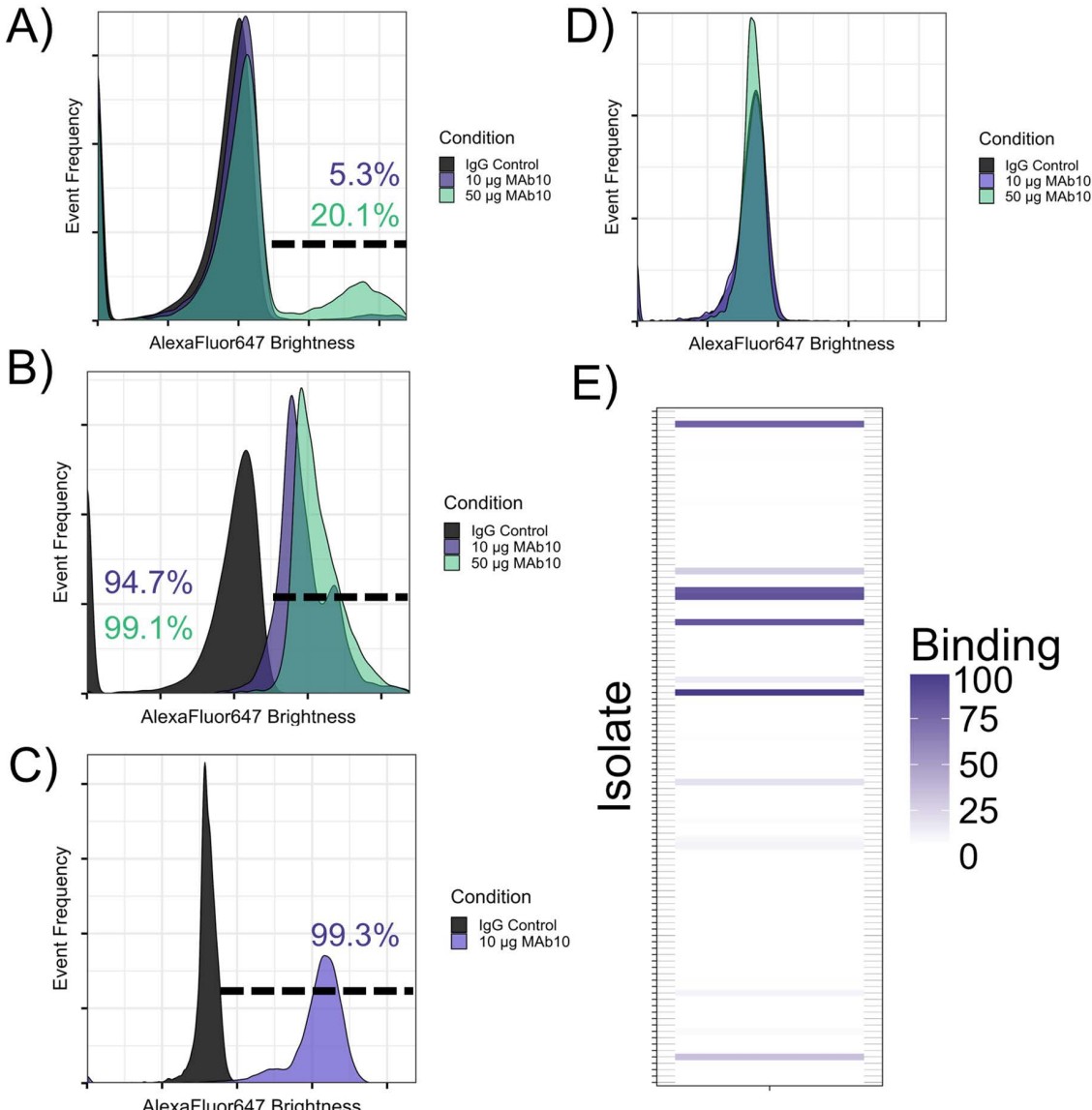

**Fig 1. MAb10 exhibits robust binding to *P. aeruginosa* isolates.** A-C) Serotype O7 isolate 1224 **(A)**, a serotype O9 clinical isolate **(B)**, or serotype O9 isolate 33356 **(C)** were assayed for binding with Mab10 via flow cytometry using 10 or 50 μg/mL conditions. The gate shows the total percentage of events in the MAb10 treated conditions that were brighter than the IgG control treated condition, color coded to match the legend. **D)** A panel of clinical *P. aeruginosa* isolates (n = 118) was assayed via flow cytometry for binding with MAb10. Of this panel, 25 isolates (21.2%) had binding with MAb10. Coloration represents the percentage binding per isolate, defined as the percentage of events in the MAb10-treated group of the flow cytometry assay that were brighter than the control group. For the full explanation of the flow cytometry gating strategy, and how percentage binding is calculated, see S1 Fig.

## Materials and methods

### Ethics statement

All animal work was conducted following approval by the Institutional Animal Care and Use Committee at the University of Southern California, in compliance with the recommendations in the Guide for the Care and Use of Laboratory Animals of the National Institutes of Health. Infected mice develop weight loss, ruffled fur, poor appetite, decreased ambulation,

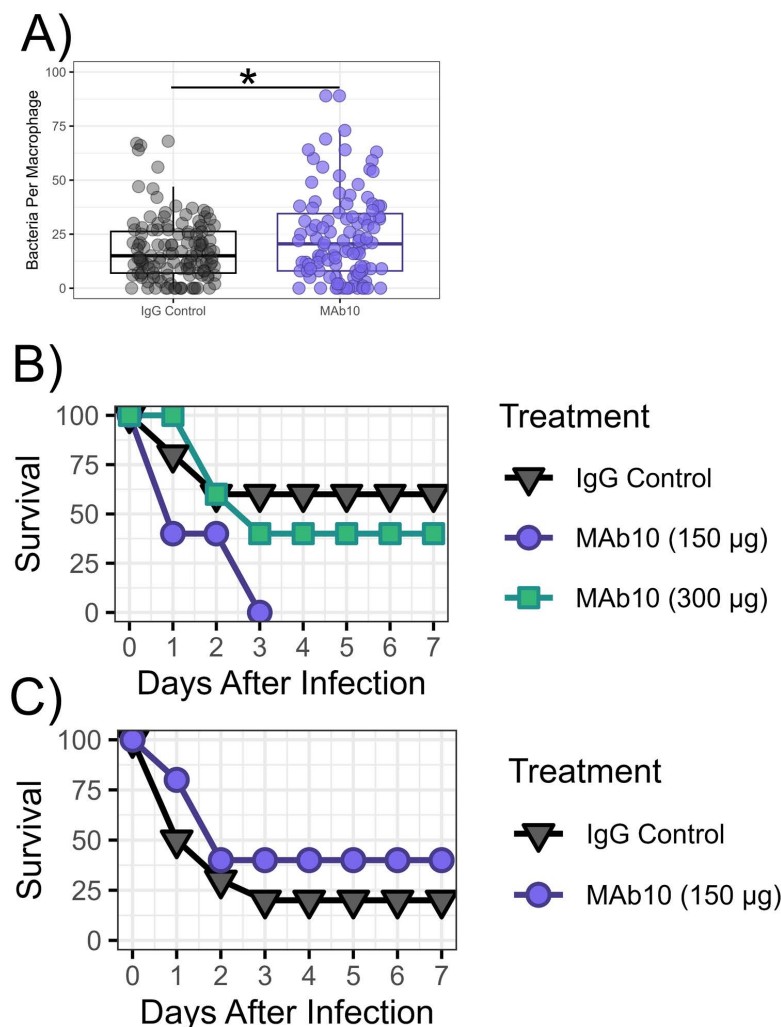

**Fig 2. MAb10 did not display *in vivo* efficacy against *P. aeruginosa* in bloodstream or pulmonary infections. A)** RAW 264.7 cells were infected with *P. aeruginosa* strain 33356 and treated with 10 μg MAb10 or isotype control antibody, with 4 technical replicates per group. Cells were fixed and stained, and bacterial cells taken up by the macrophage were counted (n = 104 cells for the MAb10 treated group, n = 128 cells for the control). SD = 14.6 for IgG control, 20.5 for MAb10 group, p = 0.04761, effect size = −0.1504275 (Cliff's Delta). **B)** Male C3H mice (n = 5) were delivered a bloodstream infection with *P. aeruginosa* strain 33356 and treated with 300 or 150 μg MAb10 or isotype control p = 0.746 and 0.095 respectively. **C)** Male C3H mice (n = 5) were delivered pulmonary infection with *P. aeruginosa* strain 33356 and treated with 150 μg MAb10 or isotype control. p = 0.24.

huddling behavior, and low body temperature. Mice that displayed huddling behavior and poor mobility were weighed 1x daily. Weight loss of greater than 15% body weight triggered immediate euthanasia via carbon dioxide asphyxiation as regulated by IACUC. Mice were monitored at least twice daily for seven days, and no animals died before meeting criteria for euthanasia. Soft bedding and other enrichment devices were provided as recommended by the veterinary staff. Nutritional supplements, such as hydrogel packs were provided as needed.

### Generation of MAb10

MAb10 was generated as previously published [9]. In brief, BALB/c mice were immunized with a sublethal inoculum (2e5 CFU/mouse) of 30 mixed *A. baumannii* isolates. Hybridomas were generated and selected as previously described [4–8].

Media was then assayed for presence of antibodies via flow cytometry for reactivity with *A. baumannii* isolates. MAb10 was purified from hybridoma media by protein G affinity chromatography and humanized and expressed as an IgG1 by Absolute Antibody as previously described into a final buffer of PBS, with endotoxin <1 EU/mg [8].

### Bacterial inoculum preparation

Bacteria were cultured overnight in tryptic soy broth (TSB) at 37°C with shaking at 200 RPM. Subcultures were diluted at 1:100 ratio and cultured for 3 hours until log phase under identical conditions, then washed 3x with phosphate buffered saline (PBS) and adjusted to the appropriate concentration for infection.

### Flow cytometry

Bacterial isolates were cultured overnight in TSB, then a 1:100 dilution subculture was incubated in TSB for 3 hours at 37°C. The cultures were washed 3x in PBS and adjusted to 1e7 CFU/mL via optical density measurement, then incubated with 10 or 50 μg/mL MAb10 or IgG control, followed by 2 μg/mL secondary antibody with AlexaFluor 647 (Thermo Fisher Scientific catalog A21445). Samples were washed twice in PBS, then flow cytometry was performed using an Accuri C6 Plus (BD). 10,000 events were collected for each sample under all conditions.

### Calculation of percent binding

Flow cytometry was done as described above using either standard or clinical method, using either an IgG isotype control antibody (Thermo Fisher Scientific, catalog MAB002, clone 11711) or MAb10, with an Alexa fluor 647 conjugated secondary antibody providing fluorescence. Using FlowJo software, events were gated for brightness with the low end of the gate set to include the brightest 1% of events in the IgG isotype control treated condition, and the high end set to the maximum brightness possible to detect. Percent binding was calculated as the percent of events in the MAb10 treated group that fall within that gate and are therefore brighter than the control group. Each strain assayed was gated individually. S1 Fig provides a clear example of the gating strategy.

### Macrophage opsonophagocytosis assay

This assay was done as previously described [25]. Briefly, RAW 264.7 macrophages ($5 \times 10^5$/well; ATCC) were cultured on a glass microscope slide coverslip and stimulated with 100 U/mL IFN-γ overnight at 37°C with 5% $CO_2$. *A. baumannii* overnight cultures were subcultured to log phase, resuspended in Hanks' balanced salt solution (HBSS) to 2e8 CFU/mL. Bacteria were added to wells at a ratio of 20:1 (bacteria to macrophages) with 10% CD-1 (IMSCD1-COMPL; Innovative Research Inc.) mouse serum and 10 μg/mL MAb10 or control antibody (Thermo Fisher Scientific, catalog MAB002, clone 11711). Macrophages fixed with 100% methanol and stained with Hema stain according to the manufacturer's protocol (Thermo Fisher Scientific). Coverslips were imaged on a Leica DMLS clinical microscope with a Leica ICC50 HD digital camera, and engulfed bacteria were counted.

### Mouse intravenous infection model

BAlb/C mice were purchased from The Jackson Laboratory (strain 000651) between 7 and 8 weeks of age at the time of infection. Mice were infected intravenously via the tail vein, followed by immediate administration of MAb10 or control treatment via the tail vein. Survival time was monitored for 7 days.

### Mouse pulmonary infection model

BAlb/C mice from The Jackson Laboratory, aged 7–8 weeks were sedated via isoflurane inhalation and suspended vertically. 50 μL bacterial inoculum was pipetted into the back of the throat, with the tongue restrained to prevent swallowing until the full inoculum was inhaled [26]. Mice were monitored for 7 days post-infection.

## Statistics

Mouse survival curves were compared using the log-rank test (α = 0.05). Macrophage phagocytosis results were compared using the Mann-Whitney U test.

## Supporting information

**S1 Fig. Bacterial flow cytometry gating strategy and acquisition of percent binding via FlowJo software.** Flow cytometry events from the IgG isotype control treated group were first viewed on a forward and side scatter plot. Excessively large or granular events were eliminated via box gate (A). Events selected via gating in figure A were then plotted as a histogram with the X axis displaying brightness on the flow cytometer's APC channel. A gate was used to select any events brighter than the 99th percentile of the IgG isotype control (B). This gate was then applied to the MAb treated group, and the percentage of events that fell within that gate were interpreted as the 'percent binding'. In the sample shown, the percent binding is 99.2% (C).
(TIF)

**S1 File. Data release.** All data addressed in this manuscript is available in the supplemental file Data_release.xlsx, with individual tabs containing data for specific figures. Data is formatted for easy analysis via R software.
(XLSX)

## Acknowledgments

Special thank you to Antonio DiGiandomenico and Joseph Horzempa for providing some of the P. aeruginosa isolates used in this work.

## Author contributions

**Conceptualization:** Matthew Slarve.

**Data curation:** Matthew Slarve.

**Formal analysis:** Matthew Slarve.

**Funding acquisition:** Brian Luna, Brad Spellberg.

**Investigation:** Matthew Slarve, Hadley Jaramillo.

**Methodology:** Matthew Slarve, Hadley Jaramillo.

**Software:** Matthew Slarve, Brian Luna.

**Supervision:** Matthew Slarve, Brian Luna, Brad Spellberg.

**Validation:** Matthew Slarve.

**Visualization:** Matthew Slarve.

**Writing – original draft:** Matthew Slarve.

**Writing – review & editing:** Brian Luna, Brad Spellberg.

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
