## [Decision Letter · Decision Letter 0]

12 Nov 2025

Dear Dr. Luna,

Thank you for submitting your manuscript to PLOS ONE. After careful consideration, we feel that it has merit but does not fully meet PLOS ONE’s publication criteria as it currently stands. Therefore, we invite you to submit a revised version of the manuscript that addresses the points raised during the review process.

We look forward to receiving your revised manuscript.

Kind regards,

Satish kumar Rajasekharan

Academic Editor

PLOS ONE

Journal Requirements:

2. Please expand the acronym “NIAID” (as indicated in your financial disclosure) so that it states the name of your funders in full.

NO authors have competing interests

Enter: The authors have declared that no competing interests exist.

5. We notice that your supplementary figures are uploaded with the file type 'Figure'. Please amend the file type to 'Supporting Information'. Please ensure that each Supporting Information file has a legend listed in the manuscript after the references list.

Reviewers' comments:

Reviewer's Responses to Questions

**Comments to the Author**

1. Is the manuscript technically sound, and do the data support the conclusions?

Reviewer #1: Yes

Reviewer #2: Yes

2. Has the statistical analysis been performed appropriately and rigorously?

Reviewer #1: Yes

Reviewer #2: Yes

3. Have the authors made all data underlying the findings in their manuscript fully available?

Reviewer #1: Yes

Reviewer #2: Yes

4. Is the manuscript presented in an intelligible fashion and written in standard English?

Reviewer #1: Yes

Reviewer #2: Yes

Reviewer #1: The manuscript presents a well-designed and scientifically sound study addressing the urgent issue of antimicrobial resistance, focusing on Acinetobacter baumannii and Pseudomonas aeruginosa. The exploration of monoclonal antibodies as an alternative to small-molecule therapeutics is both timely and innovative. The study is clearly written in standard scientific English, with a logical structure and coherent flow of ideas.The authors have made all data underlying the findings fully available, demonstrating transparency and commitment to reproducibility. The inclusion of both in vitro assays and in vivo infection models strengthens the reliability of the results. Statistical analyses appear to have been performed appropriately and rigorously, supporting the validity of the conclusions.The findings are technically sound and well interpreted. The distinction between in vitro cross-reactivity and the lack of in vivo efficacy against P. aeruginosa is handled with scientific caution, and the discussion appropriately links these observations to the broader goal of developing broad-spectrum therapeutic antibodies.Overall, the manuscript is methodologically robust, well written, and contributes valuable insights into antibody-based strategies against multidrug-resistant pathogens. I recommend publication after minor editorial revisions.

Reviewer #2: The manuscript primarily reports cross-reactivity of a previously characterized antibody; however, the novelty is limited since the lack of in vivo efficacy diminishes translational impact. Authors should better justify the originality of this work relative to prior studies

The rationale for testing P. aeruginosa appears speculative. A more systematic antigenic or epitope-level comparison should precede experimental testing to justify this direction.

The central hypothesis—that MAb10 binds pseudaminic acid-containing epitopes across species—requires biochemical validation. Without structural confirmation (e.g., glycan array or ELISA with purified carbohydrates), this remains speculative. Also, authors must discuss alternative explanations for cross-reactivity, such as nonspecific binding or shared glyco-motifs unrelated to pseudaminic acid.

The in vitro binding results rely solely on flow cytometry. There is no orthogonal confirmation (e.g., ELISA, immunoblot, or surface plasmon resonance).

The use of only one or two P. aeruginosa reference strains (O7 and O9) limits generalizability. Broader strain diversity should be analyzed or statistically summarized.

The opsonophagocytosis assay shows marginal significance (p = 0.0476) and a modest biological effect. Replication and quantitative statistical reporting (effect size, SD, n) are needed.

The manuscript lacks proper negative and positive controls for cross-reactivity—e.g., known P. aeruginosa O-antigen MAb as a comparator.

Antibody purity, concentration verification, and endotoxin levels should be described since these can affect macrophage responses and animal outcomes.

There is no mention of antibody affinity or subclass, both of which influence effector function and could explain the lack of protection.

The manuscript presents a study on cross-species reactivity of an anti-A. baumannii monoclonal antibody (MAb10) against Pseudomonas aeruginosa. The topic is timely and relevant, considering the global concern over antimicrobial resistance. However, while the study demonstrates some interesting in vitro observations, the experimental depth, mechanistic exploration, and translational significance are limited. The work lacks sufficient novelty and mechanistic insights to warrant publication in its current form. This manuscript would likely require major revisions before it could meet PLOS ONE standards.

**Do you want your identity to be public for this peer review?** For information about this choice, including consent withdrawal, please see our Privacy Policy

Reviewer #1: No

Reviewer #2: **Yes: ** Chaitany Jayprakash Raorane

---

## [Author Response · Author response to Decision Letter 1]

25 Nov 2025

Reviewer #1:

1. I recommend publication after minor editorial revisions.

We thank you for your positive assessment of our work.

Reviewer #2:

1. The novelty is limited since the lack of in vivo efficacy diminishes translational impact. Authors should better justify the originality of this work relative to prior studies

a. The rational approach to the characterization of the MAb, including the shared capsule features across species, has not been previously reported. It was therefore highly encouraging that our hypothesis was correct and that a MAb that was raised against A. baumannii was also able to bind to P. aeruginosa. Though we had hoped to see in vivo efficacy of this antibody against P. aeruginosa that was comparable to the extensive efficacy that was previously demonstrated against A. baumannii, we feel that the lack of efficacy against P. aeruginosa makes for an important scientific observation. The differential in efficacy of the MAb in these distinct species despite similar in vitro observations between the two demonstrates the challenge of producing an antimicrobial MAb that is able to treat infections caused by multiple pathogens. This in itself is the point we aim to make with this paper.

2. The rationale for testing P. aeruginosa appears speculative. A more systematic antigenic or epitope-level comparison should precede experimental testing to justify this direction.

a. The initial hypothesis was informed by the finding that MAb10 was able to bind only to A. baumannii strains that expressed capsule types that contained pseudaminic acid residues, with great consistency across a panel of 550 clinical isolates – data that was published in the paper that initially characterized MAb10. With this in mind, testing the MAb via flow cytometry against P. aeruginosa strains that were known to express pseudaminic acid was a simple and direct experiment, the results of which naturally led to the subsequent efficacy screens. Consistent with our hypothesis, the MAb was able to bind well to P. aeruginosa in vitro.

3. The central hypothesis—that MAb10 binds pseudaminic acid-containing epitopes across species—requires biochemical validation. Without structural confirmation (e.g., glycan array or ELISA with purified carbohydrates), this remains speculative.

a. Given the lack of efficacy of MAb10 against P. aeruginosa, we do not feel that further investigation into the precise epitope that MAb10 binds on this species is warranted. Our intent is to publish the finding that MAb10 binds to both species (potentially because of their shared expression of pseudaminic acid) but only shows efficacy against one. The discussion section describes this limitation.

4. Also, authors must discuss alternative explanations for cross-reactivity, such as nonspecific binding or shared glyco-motifs unrelated to pseudaminic acid.

a. This has been added to the discussion as requested (line 92)

5. The in vitro binding results rely solely on flow cytometry. There is no orthogonal confirmation (e.g., ELISA, immunoblot, or surface plasmon resonance).

a. Given the lack of translational impact of MAb10 against P. aeruginosa, we do not feel that such experiments are warranted at this time.

6. The use of only one or two P. aeruginosa reference strains (O7 and O9) limits generalizability. Broader strain diversity should be analyzed or statistically summarized.

a. We have added an additional reference strain which lacks pseudaminic acid, and which MAb10 demonstrably does not bind to. Furthermore, there is a broad screen of 118 clinical isolates of P. aeruginosa included as a heatmap in figure 1E, wherein MAb10 bound to 25 of them.

7. The opsonophagocytosis assay shows marginal significance (p = 0.0476) and a modest biological effect. Replication and quantitative statistical reporting (effect size, SD, n) are needed.

a. The figure has been updated to clarify that it is representative of two experiments with similar outcomes. The quantitative and statistical reporting has been added to the figure legend as requested.

8. The manuscript lacks proper negative and positive controls for cross-reactivity—e.g., known P. aeruginosa O-antigen MAb as a comparator.

a. To our knowledge, no such positive control is presently available. A strain with a verified serotype that lacks pseudaminic acid has been added as a negative control, and is now in figure 1D.

9. Antibody purity, concentration verification, and endotoxin levels should be described since these can affect macrophage responses and animal outcomes.

There is no mention of antibody affinity or subclass, both of which influence effector function and could explain the lack of protection.

a. This information has been added to the methods section.

---

## [Decision Letter · Decision Letter 1]

29 Dec 2025

A monoclonal antibody raised against Acinetobacter baumannii capsular carbohydrate exhibits cross-species in vitro binding against Pseudomonas aeruginosa

PONE-D-25-55530R1

Dear Dr. Luna,

We’re pleased to inform you that your manuscript has been judged scientifically suitable for publication and will be formally accepted for publication once it meets all outstanding technical requirements.

Kind regards,

Satish kumar Rajasekharan

Academic Editor

PLOS One

---

## [Editor Report · Acceptance letter]

PONE-D-25-55530R1

PLOS One

Dear Dr. Luna,

I'm pleased to inform you that your manuscript has been deemed suitable for publication in PLOS One. Congratulations! Your manuscript is now being handed over to our production team.

Kind regards,

on behalf of

Dr. Satish kumar Rajasekharan

Academic Editor

PLOS One